# Development of Cryogenic Detectors for Neutrinoless Double Beta Decay Searches with CUORE and CUPID

**Mattia Beretta** [1],* and **Lorenzo Pagnanini** [2]

1 Department of Physics, University of California Berkeley, Berkeley, CA 94720, USA
2 Department of Physics, Gran Sasso Science Institute, 67100 L'Aquila, Italy; lorenzo.pagnanini@gssi.it
* Correspondence: mattia.beretta@berkeley.edu

**Abstract:** Searching for neutrinoless double beta decay is a top priority in particle and astroparticle physics, being the most sensitive test of lepton number violation and the only suitable process to probe the Majorana nature of neutrinos. In order to increase the experimental sensitivity for this particular search, ton-scale detectors operated at nearly zero-background conditions with a low keV energy resolution at the expected signal peak are required. In this scenario, cryogenic detectors have been proven effective in addressing many of these issues simultaneously. After long technical developments, the Cryogenic Underground Observatory for Rare Events (CUORE) experiment established the possibility to operate large-scale detectors based on this technology. Parallel studies pointed out that scintillating cryogenic detectors represent a suitable upgrade for the CUORE design, directed towards higher sensitivities. In this work, we review the recent development of cryogenic detectors, starting from the state-of-the-art and outlying the path toward next-generation experiments.

**Keywords:** neutrinoless double beta decay; lepton number violation; rare event search; cryogenic detector





## 1. Introduction

Particle detectors exploit several mechanism to convert an energy deposition from ionizing radiation into a measurable quantity. The most widely used technologies measure either the ionization produced in the detector volume or scintillation signals excited in the radiation-absorbing material by the interacting particles. However, the energy detectable via these channels is only a small fraction of the total energy deposit, as the largest part is converted into lattice vibrations (i.e., heat) and escapes detection. A possible solution to increase the sensitivity consists in using a detector able to measure the vibration quanta (i.e., phonons) produced by an interaction: a cryogenic calorimeter.

In a very simplified model, a cryogenic calorimeter can be sketched as an absorber, usually a diamagnetic crystal, attached to a phonon sensor. When a particle releases energy in the absorber, its temperature increases and phonons are produced in its lattice. The phonons propagate in the absorber and are converted into a measurable signal by the sensor, acting as a thermometer. Each energy deposit in the absorber ($E_{\mathrm{Dep}}$) induces a temperature increase ($\Delta T$), given by $\Delta T = E_{\mathrm{Dep}}/C$, where C is the absorber capacity. To increase the sensitivity of this measurement method, C has to be minimized. This goal is achieved by keeping the absorber at a temperature of ∼10 mK by means of a thermal machine, usually a dilution refrigerator. In these conditions, in fact, $C \propto (T/T_{\mathrm{D}})^3$, where $T_{\mathrm{D}}$ is the Debye temperature of the material. The higher $T_{\mathrm{D}}$ is, the higher will the temperature rise due to a given energy deposit in the crystal, and thus the detector sensitivity.

The energy resolution of such detectors is their main characteristic, and it is usually below 0.1% at ∼MeV energies. This characteristic is determined by the detector technology itself, and is rather consistent with different absorbers. The resolution is only limited by the thermal noise and by eventual thermalization imperfections. Since the average energy

of a phonon is $\sim\mu eV$, the average number of phonons created by a $\sim MeV$ interaction is extremely large, allowing only small statistical fluctuation of the information carriers. Moreover, different materials can be used as absorbers, providing the possibility to investigate different processes through different strategies. The main downsides of these technique are the inability to natively measure different de-excitations, for example heat and scintillation light, and the need for high-power cooling devices to maintain the cryogenic temperatures. The former features translate into an inability to identify interacting particles producing undesired backgrounds, while the latter strongly limits the possibility to operate large detector masses with an easily scalable architecture. Cryogenic calorimeters are widely used to search for Dark Matter direct interactions by the SuperCDMS [1], EDELWEISS [2] and CRESST [3,4] collaborations, and Neutrinoless Double Beta Decay by the Cryogenic Underground Observatory for Rare Events (CUORE) [5] CUPID-0 [6], CUPID-Mo [7] and AMoRE [8] detectors.

The major running experiment based on this technique is CUORE, built with $TeO_2$ crystals to search for the Neutrinoless Double Beta Decay ($0\nu\beta\beta$) of $^{130}Te$. With respect to the native size limitation of the cryogenic calorimeters, the experience of CUORE demonstrated that these detectors can be built with masses on the ton scale, paving the way to the development of next-generation calorimetric detectors. During CUORE development, numerous efforts have been devoted to select and use radiopure materials to build the detector system, significantly reducing the level of background. At this stage, the CUORE limiting factor on this side is the degraded $\alpha$-particles background, which can be reduced only to a certain extent. To overcome this limitation, hybrid thermal-scintillation detectors have been identified as suitable candidates. Since $\alpha$ and $\beta$ particles present different scintillation mechanism, the access to a scintillation light readout allows an efficient background tagging. Such design has been carefully investigated and put to test in recent years, leading to the definition of a new path towards a next-generation cryogenic detector for the search of neutrinoless double beta decay: CUPID (CUORE Upgrade with Particle IDentification).

In this work, the parameters of interest for the $0\nu\beta\beta$ search are briefly described (Section 2) to motivate the choice of the cryogenic calorimeter approach. Subsequently, the steps leading to the CUORE design will be described (Section 3), together with a discussion of the most recent developments of cryogenic calorimeter techniques directed to the next generation of experiments (Section 4). A few alternatives to the CUORE approach will also be presented (Section 4.2), to provide a wider picture of the problem.

## 2. The Experimental Search for Neutrinoless Double Beta Decay

The incontrovertible evidence of massive neutrinos has been given by the measurement of their flavor oscillations [9–12]. These results demonstrate that the electroweak sector of the standard model is incomplete and that new physics is necessary to correctly model the observed phenomena. In this landscape, a unique role is played by the $0\nu\beta\beta$. $0\nu\beta\beta$ is a special case of the more general double beta decay ($\beta\beta$) of even–even nuclei. This process is a rare nuclear transition in which an initial nucleus (A, Z) decays to a member (A,Z + 2) of the same isobaric multiplet with the simultaneous emission of two electrons. For a few nuclei (i.e., $^{64}Zn$ [13]), this process can also have the (A, Z−2) final state, reached with the simultaneous emission of two positrons or by the means of double electron capture. According to the standard model (SM), $\beta\beta$ must obey lepton number conservation. As a consequence, the SM final state comprehends two antineutrinos and is labelled as a two-neutrino double beta decay ($2\nu\beta\beta$). In this process, the two emitted electrons carry a fraction of the total energy of the transition, called Q-value ($Q_{\beta\beta}$). Such a decay mode is shared by all of the $\beta\beta$ candidate isotopes, is characterized by half-lives $\geq 10^{18}$ years and has been observed for a dozen nuclei despite its rarity. On the other hand, assuming that neutrinos have to be explained by going beyond the SM phenomenology, $\beta\beta$ can happen with only electrons in the final state, in the so-called $0\nu\beta\beta$ mode. Being an exotic variant of an already rare process, $0\nu\beta\beta$ has expected lifetimes longer than $10^{27}$ years.

This extremely rare process is of great interest since, if detected, it would simultaneously prove the existence of lepton number-violating processes, state the Majorana nature of neutrinos and provide a value to their absolute mass scale [14–16].

The experimental search for $0\nu\beta\beta$ relies on the detection of the two electrons emitted in the process. Since the recoil energy of the nucleus is negligible, the two electrons carry a total kinetic energy equal to $Q_{\beta\beta}$. The expected signature is therefore a peak in the sum energy spectrum of the electrons centered around the $Q_{\beta\beta}$. The energy region for this search is usually referred to as the region of interest (ROI), and its width is a multiple of the full width at half maximum (FWHM) energy resolution of the used detector, evaluated at the $Q_{\beta\beta}$. The peak to be searched has to be distinguished from two main background categories. On the one hand, spurious events due to radioactivity and cosmic radiation can hide the peak. On the other, the $2\nu\beta\beta$ continuous spectrum smeared by the experimental resolution can leak in the ROI. The latter background category is unavoidable, since it comes from the $0\nu\beta\beta$ source itself. The possibility to detect the $0\nu\beta\beta$ signal depends on different detector parameters. In particular, the FWHM energy resolution ($\Delta$) and the background level ($B$) in the ROI play a crucial role in determining the effectiveness of a given experimental strategy. The performances of a given $0\nu\beta\beta$ detector are parameterized with the experimental sensitivity, $F_D^{0\nu}$, defined as the process half-life corresponding to the maximum signal that can be observed at a given statistical confidence level (CL). At a $1\sigma$ level this is given by:

$$F_D^{0\nu} \ = \ ln2\frac{\eta\epsilon N_{Av}}{A}\sqrt{\frac{T_{\mathrm{m}}M}{B\,\Delta}} \ (68\% \ \mathrm{C.L.}) \tag{1}$$

where $\eta$ is the isotopic abundance of the candidate isotope, $\epsilon$ is the detection efficiency, $N_{Av}$ is the Avogadro number, $A$ is the mass number, $T_{\mathrm{m}}$ is the measurement time and $M$ is the total detector mass. The expression in Equation (1) holds when the total background level is not compatible with zero. Such a condition is met if $M \times T_{\mathrm{m}} \times B \times \Delta > 1$, that is to say when the expected number of events during the experiment measurement time is higher than one [17]. When the experiment parameters do not satisfy this boundary, the sensitivity is better represented assuming $B \sim 0$, resulting in the following expression:

$$F_D^{0\nu}(ZB) \ = \ ln2\frac{\eta\epsilon N_{Av}}{A}\frac{T_{\mathrm{m}}M}{n_L} \tag{2}$$

where $n_L$ is a constant depending on the chosen confidence level and on the actual number of observed events. In this condition, sensitivity is directly proportional to time and mass, making it possible to scale up the experiment's reach by either increasing the mass or extending the measurement period [14].

Equations (1) and (2) summarize efficiently the most important design criteria for $0\nu\beta\beta$ detectors:

- Minimization of the continuous background (i.e., lowering $B$ factor), achievable by :
    - Placing the experiment underground, in order to reduce the cosmic ray contribution to the experimental background;
    - Building the detector with radiopure materials, in order to minimize the $\gamma$ and $\beta$ contribution to the background;
    - Cleaning the surface of materials from radioactive contaminations, in order to reduce the degraded $\alpha$ particle contribution to the background;
    - Shielding of the detector's active volume with lead and copper layers, in order to reduce the external and setup radioactivity;
    - Developing particle identification techniques, in order to discriminate degraded $\alpha$ particles from electrons signals;
    - Choosing a $0\nu\beta\beta$ candidate isotope with high $Q_{\beta\beta}$, in order to reduce $\beta$ and $\gamma$ events from the ROI and enhance the $0\nu\beta\beta/2\nu\beta\beta$ signal ratio;
- Observation of large isotope mass (i.e., increasing $M$ factor), achievable by:
    - Choosing a $0\nu\beta\beta$ candidate isotope with high natural isotopic abundance (high $\eta$);

- Choosing a detector technology that allows easy mass scalability, that is, to have high masses without substantial technological issues;

- Achieve good energy resolution (i.e., decreasing the $\Delta$ factor), to reduce the $2\nu\beta\beta$ background in the ROI;
- Long time of observation (i.e., increasing the $T_m$ factor), achievable by choosing stable detectors with low maintenance issues.

In this complex optimization problem, cryogenic calorimeters can play an important role, given their native characteristics [18]. During the last decades, these particle detectors have proven successful in trying to address some of the key sensitivity issues, but no technology exists that is able to simultaneously fulfill all requirements.

Other possibilities in solving this problem have been proposed. The most stringent limits to the half-life of $0\nu\beta\beta$ have been achieved by exploiting detectors where the $\beta\beta$ candidate is contained in the active volume, while remaining sensitive to the total energy of the emitted electrons. This strategy is characterized by high electron detection efficiency, because the source is a part of the detector, and allows the observation of large masses of the candidate isotope. Within this family, the optimization privileged some of the aspects in order to increase the attainable sensitivity. Experiments based on Germanium detectors [19–21] chose to invest mainly in superior energy resolution and active techniques for background rejection, but were limited to a single isotope of interest, $^{76}$Ge, and in achievable mass. On the other side of the spectrum, experiments based on liquid scintillators [22–24] favor the possibility to investigate more isotopes with the same setup, using extreme masses and exploiting active rejection techniques for background mitigation. Such approaches are, however, characterized by sub-performing energy resolutions, resulting in limitations due to the unavoidable $2\nu\beta\beta$ background.

## 3. The CUORE Development

The CUORE experiment history, starting from its first prototype MiBETA, is an excellent summary of the efforts needed to optimize the cryogenic calorimeter approach to the search for $0\nu\beta\beta$ [25].

Initially, the main difficulty to be addressed was increasing the reliability of these detectors in terms of both operation time and number of simultaneously operated channels. To minimize the efforts needed, $^{130}$Te was chosen as the isotope under study, given its high natural isotopic abundance (34.2% [26]). The MiBETA R&D allowed to operate initially a single crystal of $TeO_2$ for one year [27], proving the feasibility of this technique. After this initial proof of concept, a more complex array was designed and put to test, combining 20 crystals with different dimensions and mass of about 340 g [28]. These steps paved the way to the MiBETA experiment [29], which solidified the previous results and promoted the possibility to operate a large array of cryogenic detectors. At this stage, background became an issue to be dealt with in the experimental optimization. The $2\nu\beta\beta$-induced background was strongly limited by the good energy resolution of these detectors, while the spurious background had to be addressed. To reduce cosmogenic background, these experiments were placed underground in the Gran Sasso National Laboratories (LNGS), an infrastructure providing a 3600 m.w.e. shielding against cosmic rays [30–33]. From the experimental results, the main spurious background contribution was identified in the degraded alphas. The signature of these events is a continuous spectrum below 9 MeV, unavoidable in experiments sensitive only to the energy deposition. In order to reduce this contribution as well as the general level of radioactivity, the materials building the detector and those directly facing it were selected trough strict screening procedures. The goal was to identify the least contaminated materials, which were then chosen to build the future experiments. In particular, oxygen-free copper (OFC) was selected for the detector holder, and polytetrafluoroethylene (PTFE) was used to hold the crystals in place. To directly address the issue of degraded alphas, the surface of all materials was treated to remove the external layers, which are responsible for the majority of these events. Additionally, the detector shields have been optimized and chosen as a dual

copper/lead layer. To reduce the bremsstrahlung emission from $\beta$ rays, characteristics of standard lead shieldings, an ancient roman lead shield was built, characterized by low [210]Pb content [34]. Once these improvements were defined, the first step toward the final CUORE design was the CUORICINO array [35], simultaneously operating 62 crystals with different geometries. The results obtained with this experiment [36,37] allowed to select a final shape and configuration for the detector. The chosen design was based on $5 \times 5 \times 5$ cm$^3$ cubical TeO$_2$ crystals arranged in towers composed of 13 planes combining four crystals each. The design was finalized and operated as the CUORE-0 experiment [38], which served as a general test for the CUORE assembly line. The CUORE-0 detector is, in fact, one of the 19 towers now operating in the CUORE cryostat. Its operation allowed to test and finalize both the hardware and software tools needed for the ongoing CUORE operation and analysis [39,40].

Parallel to the optimization of a multi-crystal array of cryogenic calorimeters, significant steps have been made towards the development of a cryostat. The target device had to keep ∼1 ton of crystals cooled around 10 mK, while keeping at a minimum the amount of radioactive contaminants in its building materials. The design process led to the realization of the CUORE cryostat, a one-of-its-kind machine built to satisfy these impressive goals [41]. Among different optimizations, the aforementioned material selection and treatment allowed to minimize the possible backgrounds of the experiment.

The outcome of this two decade-long optimization process is the now running CUORE experiment, which established the most competitive limit on [130]Te $0\nu\beta\beta$ half-life: $T_{1/2}^{0\nu} < 3.2 \times 10^{25}$ yr [5].

## 4. Beyond the CUORE Sensitivity

The technical results obtained by CUORE showed that it is possible to run a ton-scale bolometric experiment, overcoming the mass scalability limitation related to the use of a dilution refrigerator. The next-generation experiments aim to explore the whole inverted hierarchy of neutrino masses and a relevant fraction of the normal one, reaching a sensitivity of $>10^{27}$ yr on the $0\nu\beta\beta$ half-life [26,42,43]. To achieve this goal with cryogenic calorimeters, the CUORE background level must be reduced by two orders of magnitude, exploiting a particle identification technique to discriminate degraded $\alpha$ particles from electron signals.

In the search for a novel approach, the scintillation-based detectors have proven to be effective in addressing this issue. The combination of these two techniques, resulting in scintillating bolometric detectors, was proposed in 1989 for solar neutrino experiments [44]. The first application to $0\nu\beta\beta$ searches with $\alpha$ background suppression were performed with crystal facing standard light detectors (LD), such as photodiodes [45], and were limited by the technical difficulties in using standard LDs at cryogenic temperatures. An innovation in this field was the proposal to use cryogenic calorimeters as custom low-temperature LDs [46]. This idea led to the development of new possible detector designs [47]. The main advantages of this approach is the sensitivity shown by these calorimeters. With respect to standard LDs, these devices have higher photon detection efficiencies in a wide region of photon wavelength. On the other hand, they cannot be applied in single-photon counting, due to their low native gain, and are characterized by slower response time with respect to standard photo-detectors (ms versus ns or even less). Different studies have been performed exploring the possibilities of this technique [48–50], proving the background rejection capabilities and the possibility to investigate different isotopes. This success paved the way for the construction of two demonstrators, which allowed to test different crystals and to set limits of $0\nu\beta\beta$ half-lives for two isotopes: CUPID-0 ([82]Se) and CUPID-Mo ([100]Mo ).

The excellent performance of CUORE, CUPID-0 [51,52] and CUPID-Mo [53,54] laid the foundation for CUORE Upgrade with Particle Identification, i.e., CUPID [55,56]. The main goal of the CUPID project is the design and operation of a ton-scale detector to reach a sensitivity of $10^{27}$ yr on $0\nu\beta\beta$ [26].

The progress in the development of cryogenic calorimeters is summarized in Figure 1, where we show for each experiment the exposure and the ROI background integrated in a detector FWHM, both corrected for the efficiency and expressed in unit of moles of the isotope under study [17]. These variables are calculated using the data in Table 1, extracted from the latest published data from the experiments. The evolution towards better performances of the CUORE precursors shows the results of the technical developments performed during the CUORE design optimization. The subsequent experiment kept increasing their mass, while continuously lowering the intrinsic background levels. Unfortunately, both the $^{130}$Te characteristics and the unavoidable degraded $\alpha$ background prevented this technical approach from reaching the zero-background regime. Upgrading the CUORE design using scintillating calorimeters with dual heat–light readout will allow reaching this goal. The pilot experiments (CUPID-0 and CUPID-Mo) were able to cross this boundary, reaching world-leading performances even with small exposures. The ongoing CUPID program is the next step towards the scale-up of the scintillating calorimetric technique to the CUORE scale.

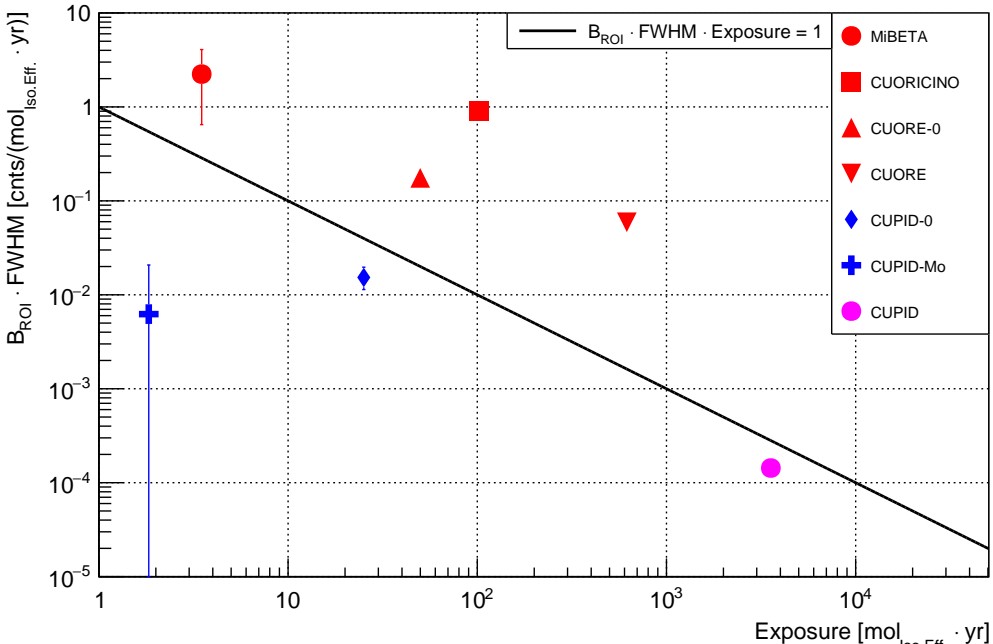

**Figure 1.** Comparison of the performances of the different $0\nu\beta\beta$ experiments based on cryogenic calorimeters described in this work. The x- and y-axes report the exposure and the ROI background integrated in a single detector FWHM, respectively. These values are both corrected for the detection efficiency and expressed in terms of moles of the $0\nu\beta\beta$ isotope under study. The variables are defined according to [17], and are calculated using the references reported in Table 1. The black line represents the boundary between the non-zero-background regime (upper side) and the zero-background regime (lower side). The cluster of red points refers to the development history of Cryogenic Underground Observatory for Rare Events (CUORE). The steps made towards reducing the background and increasing sensitive exposure are clearly visible. It is also clear that the developed technical solution is not enough to reach the zero-background region, where the increase in sensitive exposure obtained with CUORE would be more effective. It has to be pointed out that CUORE is the only running experiment in this collection, and therefore its position in this plane is expected to change with time as more data are collected. The blue points show the position of the CUPID demonstrators (CUPID-Mo and CUPID-0). Their smaller exposures are compensated by a much lower background level, obtained thanks to the combined heat–light readout. The magenta point shows the predicted performances for CUPID, foreseen to combine the best characteristics of the previous development steps. The error bars are present on each point but are hidden by the point marker.

**Table 1.** Performance of different $0\nu\beta\beta$ experiments based on cryogenic calorimeters. The data are measured for all of the experiments except for CUPID, where the best prediction of performances at the end of the operation time was reported, as derived from Monte Carlo simulations. CUORE data reflect the latest published results, and therefore these performances are expected to change as its data taking progresses.

| Experiment | Exposure$_{\text{Iso}}$ [kg yr] | Resolution [keV] | Background [cnts/(kev kg yr)] | Efficiency [%] | Reference |
|---|---|---|---|---|---|
| MiBETA | 0.66 | $8 \pm 1$ | $0.5^{+0.4}_{-0.3}$ | 84.5 | [29] |
| CUORICINO | 19.75 | $6 \pm 0.5$ | 0.2 | $82.8 \pm 1.1$ | [36] |
| CUORE-0 | 9.8 | $5.1 \pm 0.3$ | $(5.8 \pm 0.4_{\text{stat.}} \pm 0.2_{\text{syst.}}) \times 10^{-2}$ | $81.3 \pm 0.6$ | [39] |
| CUORE | 103.6 | $7.0 \pm 0.4$ | $(1.38 \pm 0.07) \times 10^{-2}$ | $77.3 \pm 0.1$ | [5] |
| CUPID-0 | 5.09 | $20.0 \pm 0.3$ | $3.5^{+1}_{-0.9} \cdot 10^{-3}$ | $70 \pm 1$ | [52] |
| CUPID-Mo | 0.48 | $7.7 \pm 0.7$ | $3^{+7}_{-3} \cdot 10^{-3}$ | $68 \pm 1$ | [57] |
| CUPID | 1000 | 5 | $10^{-4}$ | 64 | [42] |

### 4.1. The CUPID Challenge

The CUORE upgrade to achieve CUPID objectives passes through three fundamental steps:

- Increase the $0\nu\beta\beta$ emitters via isotopic enrichment;
- Active rejection of alphas and surface backgrounds in detector materials;
- Further reduction (compared to CUORE) in the $\gamma$ backgrounds by careful material and isotope selection and an active veto for muon-induced events.

In the last ten years, different isotopes (see Figure 2) and approaches were investigated for CUPID. Background rejection with scintillation, Cherenkov radiation, ionization, and pulse-shape discrimination were evaluated, and novel materials and sensor technologies were tested [58,59]. The activities carried out over the last years to design a next-generation $0\nu\beta\beta$ bolometric experiment are briefly summarized here, divided according to the isotope under investigation.

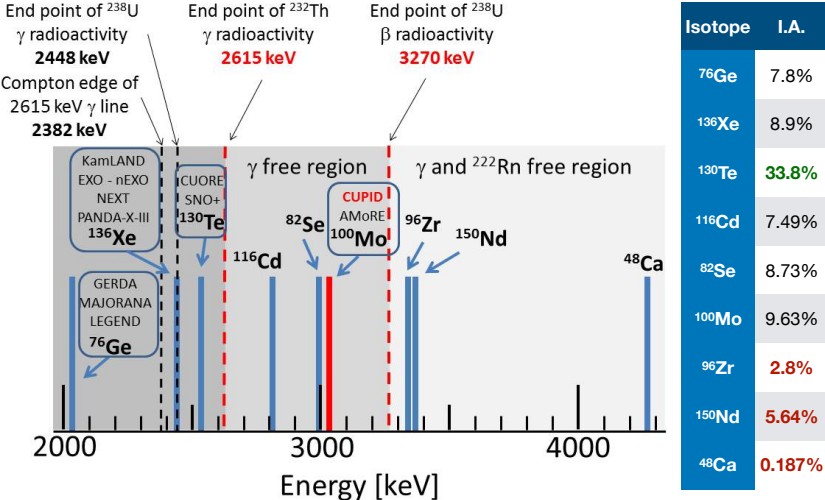

**Figure 2.** Expected energy signature in the two-electron sum-energy spectrum for the nine most suitable double-beta emitters. These are compared with background energy markers related to the maximum $\gamma$ energies of the $^{238}$U and $^{232}$Th chains and the maximum $\beta$ energy of the $^{238}$U chain. The main experiments are also mentioned in relation with their selected nucleus. For essentially technical reasons, most searches investigate the three least favorable isotopes. The isotope selected for the CUPID baseline is $^{100}$Mo. We also report the natural isotropic abundance [14]. The most favorable isotopes from the point of view of the background are also those with the lowest isotopic abundance. Figure adapted from [42].

[130]**Te.** Large-scale production of high-quality $TeO_2$ crystals was already demonstrated within the CUORE experience. However, the production of enriched crystal has to be validated to certify the energy resolution as well as the internal contaminations. This work has started on 2015, producing and operating two highly-enriched (92%) $TeO_2$ crystals (435 g each) [60]. This first test gave very satisfactory results, showing an energy resolution of 6.5 and 4.3 keV FWHM at 2615 keV, respectively. The only observable internal radioactive contamination came from [238]U (15 and 8 µBq/kg, respectively). The internal contamination of the most problematic nuclei for $0\nu\beta\beta$, [226]Ra and [228]Th, were both evaluated as <3.1 µBq/kg and <2.3 µBq/kg, respectively. Thanks to the readout of the weak Cherenkov light emitted by $\beta/\gamma$ particles by means of Germanium Neganov–Luke [61] bolometric light detectors, it was possible to perform an event-by-event identification of $\beta/\gamma$ events with a 95% acceptance level, while establishing a rejection factor of 99.99% for $\alpha$ particles. Alongside this study, silicon Neganov–Luke amplified light detectors were also successfully tested on a $1\times1\times1$ cm[3] $TeO_2$ crystal, obtaining a full rejection ($4.6\sigma$) of the $\alpha$-particles [62]. Other previous results can be found in references [63–71]. Despite the excellent results obtained, there are still two critical factors to be dealt with:

- The emitted Cherenkov light at the [130]Te $Q_{\beta\beta}$ is very small ($\sim$100 eV), and therefore the energy RMS resolution of the light detectors must be less than 20 eV, and this cannot currently be guaranteed;
- The $\gamma$-background at the [130]Te $Q_{\beta\beta}$ is one order of magnitude higher with respect to the CUPID goal.

Such limitations prevent the use of [130]Te for a next-generation experiment at the current state of the art. On the other hand, the unambiguous advantage due to the high natural isotopic abundance of [130]Te provides a good reason for keeping the optimization going on this route too.

[82]**Se.** This isotope was thoroughly investigated within the Low Background Installation For Elusive Rates (LUCIFER) program, establishing from scratch the ZnSe powder synthesis and purification protocols [72]. The crystals' growth was refined, studying the properties of dozens of ZnSe samples by means of cryogenic tests [49,73–75]. Finally 24 ZnSe crystals, 95%-enriched in [82]Se, were produced and mounted in CUPID-0 [6]. The CUPID-0 demonstrator has taken data from 2017 to 2019, demonstrating a complete rejection for $\alpha$ particles by exploiting a light-assisted particle discrimination. With this tagging capability, the delayed coincidences between [212]Bi $\alpha$-decays and [208]Tl $\beta$-decays could be identified, thus removing from the measured spectrum the [208]Tl $\beta$ events falling in the [82]Se ROI. Combining the excellent particle identification with the delayed coincidence analysis, CUPID-0 has archived a background index of $3.5^{+1}_{-0.9} \times 10^{-3}$ counts/(keV·kg· yr), the lowest value ever measured in a cryogenic calorimeter [52]. This important result established the potential of bolometers for a next-generation experiment, setting also the most stringent limit on [82]Se $0\nu\beta\beta$ half-life [52]. Finally, a full modelization of the measured spectrum has also been performed, demonstrating that the main residual background in the ROI is due to muon-induced events [76]. The deep understanding of the data also allowed a high-precision measurement of [82]Se $2\nu\beta\beta$ [77] and the extraction of limits for other rare processes [78–80]. The critical point of this technique is the mass production of high-quality ultra-pure crystals, which still demands an extensive R&D effort.

[100]**Mo.** Large-mass ($\sim$1 kg), high optical quality, radiopure [100]Mo-containing molybdate crystals have been produced and used to develop high-performance single-detector modules based on 0.2–0.4 kg scintillating calorimeters. Even if the production of such crystals was not established as $TeO_2$ ones, in a relatively short amount of time impressive performances were achieved both in term of energy resolution (4–6 keV at 3 MeV) and radiopurity (10 µBq/kg of [232]Th and [226]Ra). Moreover, the rejection of the $\alpha$-induced dominant background above 2.6 MeV is better than $8\sigma$[81,82]. An extensive R&D has been carried out to select the best candidate crystal for a next-generation experiment in the framework of the LUMINEU project and by the AMoRE collaboration [8], converging on $Li_2MoO_4$ as best candidate [83,84]. The most advanced example of this technology

is the CUPID-Mo experiment. It consists of a 2.6 kg array of 20 enriched $Li_2MoO_4$ crystals, arranged in four towers, operated as cryogenic calorimeters for more than one year. Bolometric Ge light detectors are interleaved between the crystals, resulting in all but the top elements of the towers facing an LD both at the top and the bottom [7]. In addition, the CROSS project (Cryogenic Rare-event Observatory with Surface Sensitivity) [85] is intended to develop a cryogenic detector for investigating the double beta decay of [100]Mo and [130]Te. The basic concept of CROSS consists in rejecting the background from the crystal surface by pulse-shape discrimination, assisted by a proper coating of the faces of the crystal containing the isotope of interest and serving as an energy absorber of the bolometric detector [86]. These techniques will have a major impact on background reduction in next generation experiments.

The experiences and information acquired carrying out the aforementioned activities allowed to identify enriched $Li_2MoO_4$ crystals as the best candidates for the next-generation CUPID experiment, given their excellent performances in terms of energy resolution, radiopurity and particle identification. Based on the available information, a design for the CUPID detector has been defined [42]. The different specifications are listed in Table 2. The predicted performances of the experiment have been conservatively estimated taking into account all of the information acquired during the investigation of different approaches and mainly by the CUORE, CUPID-0 and CUPID-Mo experiments. One of the main constraint of this design is the requirement for CUPID to be fully compatible with the existing CUORE infrastructure in terms of mechanical coupling, cryogenics, readout, and DAQ features. This characteristic is mandatory to boost the time schedule and to have the experiment running before 2030.

In the current design, the background index reduction is limited by the non-resolved fraction of the two-neutrino double-beta decay pile-up on the rising edge of the bolometric signal. Such an effect is due to the relatively short [100]Mo $2\nu\beta\beta$ half-life, combined with its high isotopic abundance, and the slow response time of the bolometric detectors. New methods are being developed in order to study and tackle this issue [87], and a further improvement of the pile-up rejection is expected to be achieved by combining a faster sensor (100 μs) with new analysis techniques based on Principal Component Analysis and Single Value Decomposition [58,88,89]. The exact evaluation of the expected pile-up rejection efficiency is underway and will be presented in future works of the CUPID collaboration.

**Table 2.** Main parameters of the conservative baseline CUPID detector design. The values are taken from [42].

| Parameter | Baseline |
|---|---|
| Crystal | $Li_2^{100}MoO_4$ |
| Crystal size | $4.5 \times 4.5 \times 4.5$ cm$^3$ |
| Crystal mass (g) | 241 |
| Number of crystals | 1500 |
| Number of light detectors | 1500 |
| Detector mass (kg) | 362 |
| [100]Mo mass (kg) | 200 |
| Energy resolution FWHM (keV) | 5 |
| Background index (counts/(keV$\times$kg$\times$ year)) | $10^{-4}$ |
| Containment efficiency | 79% |
| Selection efficiency | 90% |
| Lifetime | 10 years |
| Half-life limit sensitivity (90%) C.L. | $1.5 \times 10^{27}$ years |
| Half-life discovery sensitivity ($3\sigma$) | $1.1 \times 10^{27}$ years |
| $m_{\beta\beta}$ limit sensitivity (90%) C.L. | 10–17 meV |
| $m_{\beta\beta}$ discovery sensitivity ($3\sigma$) | 12–20 meV |

### 4.2. An Alternative to Cryogenic Calorimeters

As mentioned above, the scintillation-based detectors are suitable candidates for the search of $0\nu\beta\beta$. They can natively perform active background reduction and are easier to scale up than cryogenic calorimeters, since they can be operated at less extreme temperatures. In addition, these detectors are expected to allow versatility in the choice of isotope under study, good detection efficiencies, and possibility to include enriched materials. The limit to this technology is the energy resolution achievable by the light signal. This limit comes from the statistical fluctuation in the number of collected photons, limited by the light collection efficiency of the chosen scintillator-photodetector system. Such a limit can be overcome by defining a new design for performing scintillation detectors. Such a development is centered on the improvement in resolution, and has the target of FWHM $\leq 0.02 \cdot Q_{\beta\beta}$, to ensure at least a limited effect ($\leq 10\%$) of $2\nu\beta\beta$ on the background. For the chosen design, suitable scintillators containing $0\nu\beta\beta$ candidate isotopes have to be chosen. In such a context, suitable stands for the highest possible light yield and fast scintillation time. With such characteristics, the detector would not be limited by the crystal properties, allowing major improvements based on readout system upgrades. Such improvements will be related to the increase in light emission, optimization of photon extraction and optimization of readout chain. For the increase in light emission the operation at cryogenic temperatures provides an effective strategy. When the vibrational modes of a scintillating material are deactivated, in fact, the vibrational non-radiative de-excitation is damped. As a consequence, the light emission increases [90]. On photon extraction, different wrapping strategies can be implemented, surrounding the crystal with materials capable of redirecting the outgoing scintillation photons toward the photodetector. Lastly, the choice of an efficient photon detector is the most important aspect to be considered, since at this first measurement stage the actual level of the collection efficiency is determined. As a consequence, detectors with high quantum efficiency have to be preferred, in order to obtain better performances [91]. Additionally, consideration of gain fluctuations, non-uniform efficiency and readout noise have to be considered, since all of these aspects take part in spoiling the resolution. The FLARES (Flexible Light Apparatus for Rare Events Search) project tried to follow this concept. It proposed the development of a detector based on the use of Silicon Drift Detectors (SDD) to read the light emitted by large scintillation crystals, cooled at 120 K [92,93]. SDDs are characterized by low noise and high quantum efficiency, while the scintillators containing $0\nu\beta\beta$ candidates can be enhanced in their performances by the low temperatures.

Considering a completely different approach, the resolution problem can be addressed by selecting more performing scintillating materials. In this framework, the scintillating nanocrystals play an important role. These new generation of compounds, in fact, is characterized by extremely high light emission, and they can be designed to work at a certain wavelength, thus making the optimization of their readout chain easier [94,95]. In addition, the absence of non-radiative de-excitation channels makes the radiative recombination extremely quick, with scintillation emission time on the order of few ns. At the current status, different nanoscintillators are available, in terms of different structures and constituting elements. This wide range of choices makes them versatile, since it is possible to choose the needed materials with the needed emission spectrum to fit the demands of a certain application. In particular, some of these materials can be built using elements with isotopes candidates to $0\nu\beta\beta$, such as Te or Se. Different studies have been performed with quantum dots (QDs) used as doping for liquid scintillators, since these materials provide both a wavelength shifter and a source for the $0\nu\beta\beta$ [96–98]. On the other hand, using these materials simultaneously as radiation emitters and absorbers presents more difficulties. The accumulation of great masses of nanocrystals is, in fact, problematic, since the self-absorption limits the light output. In addition, such compounds are produced as powders, needing a matrix to be arranged in more useful structures. Such difficulties prevented their application to the scintillation-based detection of radiation. In this framework, the ESQUIRE (Experiment with Scintillating QUantum dots for Ionizing Radiation Events)

project proposes a possible pathway to the development of new scintillation detectors, aimed to a final application to the search of $0\nu\beta\beta$ [99]. ESQUIRE R&D work focuses on the selection of different QDs with optical characterization, followed by a direct scintillation measurement performed with highly sensitive solid-state photon detectors. The initial work allowed to demonstrate the possibility to apply nanocrystals as fast scintillators [100], validating a part of its initial goals.

## 5. Conclusions

Developing $0\nu\beta\beta$ detectors forces the design of innovative solutions to overcome the congenital difficulties of this frontier research. During the last 30 years, the application of cryogenic detectors in this field rapidly progressed thanks to continuous R&D work culminated in the CUORE construction and operation. This experiment is currently demonstrating the possibility to operate 1000 crystals as cryogenic calorimeters in a single cryostat, paving the way to a new generation of $0\nu\beta\beta$ detectors. Its performances provide a milestone in terms of energy resolution ($7.0 \pm 0.4$ keV at $^{130}$Te $Q_{\beta\beta}$) and background (($1.38 \pm 0.07$)$\cdot 10^{-2}$ cnt/(keV·kg· yr)), defining the goal that a future experiment has to fulfill—in particular, mass on the ton scale, resolution of few keV at the $Q_{\beta\beta}$ and a background of less than $10^{-2}$ cnt/(keV·kg· yr). The next step in this process will be the CUPID experiment. Its development is ongoing and relies both on the CUORE experience and on the CUPID-0 and CUPID-Mo prototypes. The new challenges to be overcome, especially the $2\nu\beta\beta$ pile-up, are currently being tackled by combining new hardware and analysis solutions to improve energy and time resolution. The long experience achieved also pointed out the issues that cannot be solved within the current CUORE infrastructure, opening the investigation of other possible solutions for a next-generation $0\nu\beta\beta$ experiment. It appears clearly then that the development of cryogenic calorimeters is a challenging task, capable to produce innovation in the radiation detector landscape.

**Author Contributions:** Conceptualization and writing, M.B. and L.P. All authors have read and agreed to the published version of the manuscript.

**Funding:** This research received no external funding.

**Institutional Review Board Statement:** Not applicable.

**Informed Consent Statement:** Not applicable.

**Conflicts of Interest:** The authors declare no conflict of interest.

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
