# Peer review of "Development of Cryogenic Detectors for Neutrinoless Double Beta Decay Searches with CUORE and CUPID"

_applsci, doi:10.3390/app11041606_

Round 1

Reviewer 1 Report

  • The title is very generic while the article mostly focuses on reviewing the recent detector developments in the framework of the CUPID / CUORE experiments. I recommend to choose a less generic title that better emphasises the scope of this review.

  • It will be helpful to the reader if the authors add a paragraph detailing how experiments based on cryogenic detectors fit in the global landscape of next-generation neutrinoless double beta decay experiments in terms of sensitivity and other key parameters (mass, energy resolution, background).

  • Technical questions to the authors (I would welcome if the authors incorporate answers to these into the main text where possible):

    • The authors emphasise the need for tonne-scale experiments throughout the text. However, CUPID will be below that number in terms of detector mass (table 2). What are the limiting factors that prevent going towards larger detector masses. How does that affect CUPID when compared to other next-generation experiments in the field and is there work being done to overcome such limitations?

    • The authors argue that cryogenic detectors should have a superior energy resolution (excitation of phonons) compared to other technologies (i.e., charge and photon based detector). However, table 1 suggests that the energy resolution of CUPID will be comparable to what is already achieved with e.g., HPGe detectors. What is limiting the energy resolution in the detectors used for CUPID? (Also, how will the improvement in energy resolution when going from CUPID-Mo to CUPID be achieved?)

  • The main text requires additional proof reading and spell checks

Author Response

We provide a point-by-point response in the attached file. 

Reviewer 2 Report

In the manuscript titled "Development Of Cryogenic Detectors For Neutrinoless Double Beta Decay Searches" a review of the 0νββ research with cryogenic detectors is presented. The work is exhaustive and describes in details the main milestones from the characterization of a single crystal to a tonne scale experiment and the main techniques to reduce background and the big efforts done to improve the detector sensitivity. In particular the choice of crystals allowing pulse shape analysis permits to reach the boundary condition of background compatible with zero and it seems this can be a real breakthrough for the goal of the measurement

The manuscript is very clear, well referenced and adequate in length and concepts. For this reason, I recommend for publication after only minor changes.

Just for completeness, a fast comparison and the list of pro and cons of the bolometric technique with respect other techniques used in the 0νββ research field (e.g. Germanium in the GERDA experiment) can be done.

Moreover, I invite the authors to consider the implementation of these suggestions to make clearer the text

line 30: can you quantify the energy resolution range of such detectors and why this is the main characteristic?

line 31: is thermalizations or thermalization? 

line 40: on THIS/THE technique

line 50: remove the final e in word "designe"

Equation 1: the symbol N_av should be specified 

line 89: briefly try to explain why the condition is met when MTBD>1, or at least insert a reference 

line 142: expand OFC and PTFE and then put the acronym in brackets

line 163: quantify the limit reached by CUORE experiment

Figure 2: if possible, please add near the isotopes also the corresponding natural abundance, in this way one gives also another parameter of comparison between different isotopes used

Table 1: do the best prediction for the CUPID experiment  come from Monte Carlo simulation? if yes, specify it if no explain how these come from. ( you can add in the text and in the table caption write "further details in the text") 

line 218: specify how highly enriched are the crystals

line 236 : rephrase the sentence in "studying the properties of dozens of ZnSe samples by means..".

line 255: please, insert a reference or quantify the rejection values of the two crystals (e.g is better then 8 sigma going from xxx to yyy)

line 276: could you estimate or there is in literature an estimation on how the pile up will be reduced ( and the total background,too) by the new method mentioned?

line 283: i think the verb misses in the sentence

Table 2: in general the discovery sensitivity is defined at to 5 sigma, isn't it? why in the table it is specified that it is 3 sigma? Or am I misunderstanding the meaning of the line in the table?

line 317: QDs --> Quantum Dots (QDs) 

Paragraph 4.6 in general: it is interesting that the authors describe a lot of future prospectives for the improvement of energy resolution at the Q value of reaction beyond CUPID. But it is not clear to me if this techniques could have some drawbacks in terms of detection efficiency or enrichment of active materials. I have the impression that the answer is no, but please specify it.

Author Response

(The authors gave the same response as above.)

Reviewer 3 Report

The article presents a concise but complete overview of the development of cryogenic detectors to investigate the neutrinoless double beta decay over three decades.
This type of detectors have released very important results for double beta decay searches and the study of neutrino properties and are also used in other areas.
The achievements in the performance of the detectors are highlighted and work in progress is also described providing and outlook of the field. Suitable references are given to go in depth in the many work lines described.
For all these reasons, I strongly support the publication of the article in Appl. Sci. I recommend a revision just to correct typos or format issues and to complete some aspects.
I list below my comments and suggestions, indicating the corresponding line at the manuscript.

5: I would complete "energy resolution at the signal peak"
5: add: have BEEN proved effective
28: add: at A temperature OF
29-30: a comment about which kind of materials have an appropriate TD would be useful
30: I would remove ,
35: correct through (missing h)
36: could you clarify "multi-messenger measurements"? Do you refer to the simultaneous measurement of heat and charge/ionization?
40: add: based in THIS technique
41: you should define the abbreviation 0nbb in this first appearance (it is made later at line 60)
48: "clean" must not be understood, it could be better to use "radiopure"
50: designe -> design
Section 1: you could include a mention to the development and use of cryogenic and hybrid detectors also in the context of other rare event searches like the direct detection of dark matter.
53-55: I suggest providing a more detailed presentation of the manuscript, indicating specifically the contents of each section of the paper.
62-63: for completeness, you could mention the possibility of the double beta decay to (A,Z-2) with double EC or beta+ processes.
65: two neutrino -> two-neutrino
67: half lives -> half-lives
66-67: it would be useful to state that the 2nbb process, although rare, has been observed for several nuclei.
70: live times -> lifetimes
71: add: IT would
72: together with Refs. [5,6] you may consider to provide a more recent review of double beta decay searches, like for instance:
Dolinski, M.J.; Poon, A.W.P.; Rodejohann, W. Neutrinoless Double-Beta Decay: Status and Prospects.
Ann. Rev. Nucl. Part. Sci. 2019, 69, 219.
https://www.annualreviews.org/doi/10.1146/annurev-nucl-101918-023407
77: you could introduce here the abbreviation FWHM, later used
79: as later explained in page 3, not only external radioactivity give spurious events at the ROI. More generally, you could say here "due to radioactivity and cosmic radiation".
87, 92: use \noindent command before where
87-88: define also Nav as Avogadro number and A as the mass number. I guess subindex mis for Tmis comes from Italian, it could be just Tm. It would be useful also to indicate if the energy resolution parameter Delta corresponds to the FWHM.
93: I would put , after condition
101; add ; at the end
109: add: SIGNAL ratio
112: I think "to say possibility" can be removed, writing just "... that is, to have..."
116: I would put , before achievable
130: a mention to the order of dimensions/mass of the MiBETA crystals would be useful
136: at least Refs. [14,15] describe not the LNGS as an underground lab but specifically HPGe detectors used for material screening. If given these references, a mention to this facility (very important to achieve low backgrounds) in the text could be done.
138: I would better write alfa's (it appears later several times more).
141: correct through (missing h)
142: define OFC
146: bremmstralhung -> bremsstrahlung
147: add: aN ancient
149: add: crystalS
160: aformentioned -> aforEmentioned
162: 2-decade -> two-decade
163: you could consider to present Figure 1 and Table 1 (which are really very interesting and useful) here in this section 3, to complete the details on the improved results during this process obtained from MiBETA, CUORICINO, CUORE-0 and CUORE.
Although the focus is on the detector performance, you could consider to give also in Table 1 the relevant physical results like the obtained half-life limit on the neutrinoless double beta decay and even the limits on the effective neutrino mass (as later made in Table 2).
168 (and 190): the half-life value quoted refers to a particular double beta decay emitter? If so, it could be stated.
174: add: application TO
176: where -> were
196-197: some word seems to have been lost in sentence "The experiment were increasingly bigger,..."
209: I would remove ,
232: despite these two critical factors, work for 130Te investigation is still ongoing? A comment on this would be in order.
283: add: technology IS the energy
288: I would put , after design
303: I would remove ,
317: define QDs abbreviation
334: add: crystalS
338: combined -> combining?
343: I suggest including in the conclusion a summary of the presently achieved values of the relevant experimental parameters and the required values for the coming experiments.

General comment: the CROSS project (Cryogenic Rare-event Observatory with Surface Sensitivity) is also intended to use cryogenic detector for investigating the double beta decay of 100Mo and 130Te.
You could also mention it for the sake of completeness and to put it in the context and compare with the described developments.

REFERENCES
[6] It seems incomplete, please give the full reference
[13] Put in capitals: Laboratori Nazionali
[15] A strange symbol appears, I guess for microBq
[23] It seems incomplete, please give the full reference
[54] Subindex lost for TeO2
[65] Space missing between of 82Kr
[69] Subindexes lost for Li2MoO4
[76] Write title in lower case
[83] Write journal name in lower case

Author Response

(The authors gave the same response as above.)
